# The Association between Inefficient Repair of DNA Double-Strand Breaks and Common Polymorphisms of the HRR and NHEJ Repair Genes in Patients with Rheumatoid Arthritis

**DOI:** 10.3390/ijms25052619

**Published:** 2024-02-23

**Authors:** Grzegorz Galita, Joanna Sarnik, Olga Brzezinska, Tomasz Budlewski, Marta Poplawska, Sebastian Sakowski, Grzegorz Dudek, Ireneusz Majsterek, Joanna Makowska, Tomasz Poplawski

**Affiliations:** 1Department of Clinical Chemistry and Biochemistry, Medical University of Lodz, 92-215 Lodz, Poland; grzegorz.galita@umed.lodz.pl (G.G.); ireneusz.majsterek@umed.lodz.pl (I.M.); 2Department of Rheumatology, Medical University of Lodz, 92-115 Lodz, Poland; joanna.sarnik@umed.lodz.pl (J.S.); olga.brzezinska@umed.lodz.pl (O.B.); tomasz.budlewski@umed.lodz.pl (T.B.); joanna.makowska@umed.lodz.pl (J.M.); 3Biobank, Department of Immunology and Allergy, Medical University of Lodz, 92-213 Lodz, Poland; marta.poplawska@umed.lodz.pl; 4Faculty of Mathematics and Computer Science, University of Lodz, 90-238 Lodz, Poland; sebastian.sakowski@wmii.uni.lodz.pl (S.S.); grzegorz.dudek@wmii.uni.lodz.pl (G.D.); 5Centre for Data Analysis, Modelling and Computational Sciences, University of Lodz, 90-128 Lodz, Poland; 6Department of Pharmaceutical Microbiology and Biochemistry, Medical University of Lodz, 92-215 Lodz, Poland

**Keywords:** rheumatoid arthritis, comet assay, DNA double-strand break, DNA repair, single-nucleotide polymorphism, RAD50, RAD51B

## Abstract

Rheumatoid arthritis (RA) is an autoimmune disease characterized by chronic inflammation affecting up to 2.0% of adults around the world. The molecular background of RA has not yet been fully elucidated, but RA is classified as a disease in which the genetic background is one of the most significant risk factors. One hallmark of RA is impaired DNA repair observed in patient-derived peripheral blood mononuclear cells (PBMCs). The aim of this study was to correlate the phenotype defined as the efficiency of DNA double-strand break (DSB) repair with the genotype limited to a single-nucleotide polymorphism (SNP) of DSB repair genes. We also analyzed the expression level of key DSB repair genes. The study population contained 45 RA patients and 45 healthy controls. We used a comet assay to study DSB repair after in vitro exposure to bleomycin in PBMCs from patients with rheumatoid arthritis. TaqMan SNP Genotyping Assays were used to determine the distribution of SNPs and the Taq Man gene expression assay was used to assess the RNA expression of DSB repair-related genes. PBMCs from patients with RA had significantly lower bleomycin-induced DNA lesion repair efficiency and we identified more subjects with inefficient DNA repair in RA compared with the control (84.5% vs. 24.4%; OR 41.4, 95% CI, 4.8–355.01). Furthermore, SNPs located within the RAD50 gene (rs1801321 and rs1801320) increased the OR to 53.5 (95% CI, 4.7–613.21) while rs963917 and rs3784099 (RAD51B) to 73.4 (95% CI, 5.3–1011.05). These results were confirmed by decision tree (DT) analysis (accuracy 0.84; precision 0.87, and specificity 0.86). We also found elevated expression of RAD51B, BRCA1, and BRCA2 in PBMCs isolated from RA patients. The findings indicated that impaired DSB repair in RA may be related to genetic variations in DSB repair genes as well as their expression levels. However, the mechanism of this relation, and whether it is direct or indirect, needs to be elucidated.

## 1. Introduction

Rheumatoid arthritis (RA) is a chronic, inflammatory, and autoimmune rheumatic disease characterized by an improper autoimmune response. RA affects about 2% of the population worldwide. The likelihood of developing RA increases with age, but the disease can develop at any age [1]. Women are much more likely to suffer from RA than men, a discrepancy that attempts to link to hormonal disorders, although there is no convincing evidence for this. Other factors related to the development of RA include environmental factors such as air pollution, lifestyle, including smoking, and genetic factors [2]. The latter are believed to be the main contributors to the development of the disease mainly associated with the human leukocyte antigen (HLA) locus; however, other genes are also involved [3]. Genome-wide association studies (GWAS) have not only identified genetic elements as risk factors for the development of RA but have also shown significant variation between ethnic groups [4]. It is generally accepted that some of the genetic factors of RA are common to the human population; nevertheless, other studies show variation [5,6,7].

The molecular basis of RA is complex. The immune system of individuals with RA produces antibodies targeting multiple proteins that undergo a variety of post-translational modifications. This heterogeneity of the antibodies suggests multiple metabolic abnormalities within peripheral blood mononuclear cells (PBMCs). These abnormalities, as we and other authors have previously demonstrated, involve, among other processes, the PBMC response to DNA damage, including DNA repair [8,9,10,11,12]. PBMCs isolated from subjects with RA are characterized by greater sensitivity to DNA damage agents than healthy subjects. Furthermore, the kinetics of repairing oxidative DNA damage and DNA double-strand breaks (DSBs) are impaired. The reason for these abnormalities in the repair of oxidative DNA damage is the altered expression and occurrence of different allelic forms of genes encoding repair proteins [13]. Changes in DSB repair are attempted to be associated with reduced expression of the key protein ATM [9]. To date, analysis of the function of DSB repair in RA has been rather superficial—carried out either on a small study sample or non-Caucasian ethnic groups and without analysis of the genetic background. Taking these facts into account, in the present study, we analyze the efficiency of DSB repair in a Caucasian population in correlation with the genotype, limited to the occurrence of single-nucleotide polymorphisms (SNPs) in the DSB repair genes. We confirmed the phenotype-genotype correlation using machine learning methods and also analyzed changes in the expression of key DSB repair genes at the mRNA level in PBMCs isolated from subjects with RA.

## 2. Results

### 2.1. Characteristics of the Study Population

There were no significant differences in the distributions of age, sex, and smoking status between cases and controls. The mean duration of the disease was 14.9 ± 14.5 years (from 1 to 75 years). Twenty-three patients were currently (for at least one month before blood collection) receiving disease modifying anti-rheumatic drugs (DMARDs), including methotrexate and/or sulfasalazine, and twenty-two patients did not receive DMARDS within the last month. Glucocorticosteroids (GCS) were used for the treatment of twenty-two patients. Nine patients did not have rheumatoid factor levels (positive in 36 cases). We detected an anticyclic citrullinated peptide antibody (aCCP) in 32 patients. Additionally, we determined the level of C-reactive protein (CRP) (16.2 ± 20.5 g/dL). Disease activity has also been also assessed based on the disease activity score 28-joint count C-reactive protein (DAS28)-CRP score (DAS <1.7 was defined as remission—3 patients, DAS > 1.7 and <2.6 was defined as low disease activity—15 patients, and DAS28 above 5.1 as high disease activity—27 patients). All controls had CRPs within normal limits and no chronic disease with inflammatory background (Table 1).

### 2.2. Differences in DNA RepEff between RA Patients and Controls

PBMCs isolated from RA patients had a significantly lower efficiency of bleomycin-induced DNA lesion repair as presented on Figure 1 (<0.0001). The median of RepEff in the control group was 78 vs. 46 calculated in the RA group. The Hodges–Lehmann estimation of the location shift was 26.33.

Moreover, we identified more subjects with marginally efficient and inefficient DNA repair in RA as compared with the controls (43 vs. 22; Table 2).

### 2.3. The Allele and Genotypes of Key BER Genes in RA Patients and Controls

The RA and control groups were almost similar in the genotype distribution of common polymorphisms located in key DNA double-strand break genes (Table 3). Using univariate logistic regression, we found an association of the two SNPs of the *RAD51* (rs1801320, rs1801321) and *RAD51B* (rs963917, rs3784099) genes with RA. Regarding the aims of this manuscript, we paid special attention to these SNPs in regard to the correlation of the phenotype (RepEff) with the genotype (SNPs).

### 2.4. Associations between RepEff (Phenotype) and DSB SNPs (Genotype) and RA

To estimate RA risk, the relative RepEff was grouped into the quartile values of the controls (Table 2). The crude ORs for RA risk associated with the relative RepEff in the second group, third group and fourth group were 1.1 (95% CI, 0.06–19.6), 5.45 (95% CI, 0.55–54.28) and 41.5 (95% CI, 4.84–355), compared with the first group (highly efficient DNA repair). After adjusting for the SNPs rs1801321, rs1801321, rs963917, and rs10483813, in the multivariate logistic regression analysis, the ORs of the RepEff increased in the fourth group corresponding to the no repair phenotype (Table 4). We have also tested possible correlations between RepEff and the clinical parameters of RA like DAS, RF, aCCP, and CRP; however, no correlations were found.

### 2.5. Differences in DSB Gene mRNA and miR-155 Expression Levels between RA Patients and Controls

We also observed the deregulation of the expression level of DSB genes in RA as opposed to the controls. A significant statistical difference in the level of gene expression was calculated for the *RAD51B*, *BRCA1*, and *BRCA2* genes. The RA group shows higher gene expression levels in *RAD51B*, *BRCA1*, and *BRCA2* than the control group (median of 0.00213 vs. 0.0069; 0.004 vs. 0.008; 0.00057 vs. 0.001; *p* < 0.05). No difference between the RA and control groups was found in the expression of *RAD51*, *ATM*, *PRKDC*, and *H2AX*; however, we noticed a lowered trend in the level of *H2AX* mRNA (0.046 vs. 0037, Figure 2).

### 2.6. Decision Tree (DT) Analysis

To identify the most important inputs for the DT classifier, which ensures the highest accuracy in recognizing RA patients, we applied a sequential feature selection procedure. It operates in two variants: forward and backward. Sequential forward feature selection (SFS) gradually adds one feature at a time to the selected subset, while sequential backward feature selection (SBS) gradually eliminates the features that contribute the least to the model’s performance from the initial set. Both variants explore different combinations of features and aim to find a subset of features that enhances the model’s predictive ability while avoiding the inclusion of irrelevant or redundant features that may introduce noise or overfitting. Because SFS and SBS employ a greedy algorithm working in opposite directions, they usually lead to different results.

Figure 3 illustrates the process of searching the feature space using SFS and SBS. Note that SFS indicated x_1 (RepEff) as the most relevant feature, achieving an accuracy of 0.8000 when used as the sole feature in the DT model. In this case, DT uses only one node and can be expressed using the following simple decision rule: if x_1 = 3 then
$\hat{y}$ = 1 else
$\hat{y}$ = 0. By adding x_10 to the model, the accuracy increases to 0.8222. The resulting decision rule becomes: if x_1 = 3 then
$\hat{y}$ = 1 else (if x_10 = 1 then
$\hat{y}$ = 1 else
$\hat{y}$ = 0). Further, adding x_11 leads to an accuracy of 0.8444. The tree for this case is depicted in the left panel of Figure 3.

The results obtained from SBS show rather low accuracy when all 30 features are included in the model (Acc = 0.6889). However, by progressively eliminating certain features (x_2, x_3, …, x_29), as illustrated in Figure 3, the accuracy gradually improves to 0.8556. Removing subsequent features, x_5, x_7, …, x_28, does not negatively impact the accuracy. Finally, the features shown in Table 5 were selected. The tree constructed using these features is shown in the right panel of Figure 4.

For comparison, Figure 5 shows the feature importance estimated using three standard methods. Chi2 examines whether each feature is independent of a response variable by using individual chi-square tests. The output score, –ln(p), is based on the p-value of the test statistic and expresses a strength of the relationship between the corresponding feature and the response variable. The minimum redundancy maximum relevance algorithm (MRMR) identifies an optimal set of features that are mutually dissimilar yet effectively represent the response variable. ReliefF feature scoring relies on detecting differences in feature values among nearest neighbor instance pairs. The algorithm penalizes features that yield different values for neighbors of the same class and rewards features with different values for neighbors of different classes. Unlike Chi2, MRMR and ReliefF are sensitive to feature interactions.

As shown in Figure 5, all three methods highlight $x_{1}$ as the most important feature, aligning with the selections made by SFS and SBS. Additionally, $x_{11}$, chosen using both SFS and SBS, receives high importance scores across all algorithms. However, $x_{22}$, which obtains a high importance score in Chi2, MRMR, and ReliefF, was not selected using the sequential methods. In general, the results obtained from Chi2, MRMR, and ReliefF lack consistency and may not be definitive. It is important to note that SFS and SBS evaluate features within the context of a specific model, whereas Chi2, MRMR, and ReliefF do not consider the predictive model in their assessments.

Table 6 provides a comparison of DT performance using different input configurations: only $x_{1}$, all available features, features selected by SFS, and features selected by SBS. The performance metrics were determined using leave-one-out cross-validation. The DT hyperparameters were determined through preliminary experiments and set as follows: split criterion—Gini’s diversity index and maximum number of splits—20. The results in Table 6 highlight the significant discriminative power of $x_{1}$ as a standalone feature. When augmented with the features selected by sequential selection algorithms, the model’s discriminative ability is further enhanced. However, it is important to note that including all available features diminishes the model’s accuracy. This clearly suggests that many of these features lack relevant information and should be eliminated from the model.

## 3. Materials and Methods

### 3.1. Study Groups

The study group included 45 patients diagnosed with RA hospitalized at the Department of Rheumatology, Medical University of Lodz, and in the outpatient clinic. A total of 45 healthy subjects without symptoms of chronic inflammatory conditions and cancer history in their closest relatives were selected as the control group. All RA patients met the European League Against Rheumatism/American College of Rheumatology (EULAR/ACR) 2010 diagnostic criteria for RA. The study was conducted according to the guidelines of the Declaration of Helsinki and approved by the Institutional Bioethics Committee of the Medical University of Lodz (Lodz, Poland) (no. RNN/07/18/KE, approved date: 16 January 2018) and informed consent was obtained from all subjects involved in the study.

### 3.2. PBMC Isolation

PBMCs were isolated from 9 mL of the peripheral blood of the study group in a density gradient using Lymphosep (Biowest, Nuaillé, France). Blood diluted 1:1 in PBS was gently applied to the Lymphosep and centrifuged for 15 min (400× *g*) at room temperature. The collected PBMCs (1 × 10^5^) were washed 2 times in PBS.

### 3.3. Comet Assay

Assessment of endogenous DNA lesions as well as DNA lesions resulting from the exposure of PBMCs to bleomycin and DNA repair effectiveness analyses were performed using alkaline single-cell gel electrophoresis (comet assay). PBMCs treated with 25 μM bleomycin 30 min at 37 °C were suspended in low melting point (LMP) agarose (Sigma-Aldrich Corp., St. Louis, MO, USA) and applied to slides coated with normal melting point NMP agarose (Sigma-Aldrich Corp., St. Louis, MO, USA). To study DNA repair, the cells were allowed to recover for 120 min in fresh medium before suspension in LMP agarose. The prepared slides were incubated in the lysis buffer at pH 10 (2.5 m NaCl, 10 mm Tris, 100 mm EDTA) with 1% TritonX-100 (Sigma-Aldrich Corp., St. Louis, MO, USA) for 1 h at 4 °C. After the incubation, slides were left in development buffer (300 mm NaOH, 1 mm EDTA) for 20 min at 4 °C. The preparations were then electrophoresed in an electrophoresis buffer (30-mMNaOH, 1 mm EDTA) under the following conditions: 17 V, 32 mA, 20 min, RT. Slides were rinsed 3× with distilled water and fluorescently stained with DAPI. The stained “comets” were analyzed using a fluorescence microscope Nikon CI-L plus (Nikon, Tokyo, Japan) with Lucia software v.6.70 (Laboratory Imaging, Prague, Czech).

The individual DNA repair efficiency was calculated as shown previously [13]. Briefly, we subtracted the percentage of DNA damage measured after 120 min of repair from the initial damage score set as 100%. Then, we set the repair ranks based on the repair efficiency quartiles of the control group. Quartile 4 means highly efficient DNA repair, quartile 3 efficient repair, quartile 2 marginally efficient DNA repair and quartile 1 no repair.

### 3.4. DNA Isolation

DNA was isolated from the peripheral blood of RA patients and controls using the GeneMatrix Blood DNA purification Kit (EURx, Gdansk, Poland). The peripheral blood of patients was lysed in the presence of a buffer containing proteinase K and chaotropic salts. Ethanol was then added to selectively bind the DNA to the membrane in the spin-column. After a short centrifugation, the DNA was bound to the membrane, while unbound impurities remained in the column effluent. In the next step, the samples were washed two times using wash buffer to remove the remaining contaminants from the membrane containing the DNA. Purified DNA was eluted with a low-salt buffer containing Tris-EDTA. The concentration and purity of the obtained DNA were evaluated spectrophotometrically by measuring absorbance at 260/280 nm on Synergy HT spectrophotometer (BioTek, Hong Kong, China).

### 3.5. Determination of SNPs

The polymorphic variant frequency of DSB repair-related genes (Table 7): *XRCC2* (rs3218536), *RAD51* (rs1801320, rs7180135, rs45549040, rs1801321, and rs2619681), *RAD51B* (rs963917, rs963918, rs3784099, and rs10483813), *TP53* (rs1042522), *RAD52* (rs1051669), *MRE11A* (rs2155209), *XRCC6* (rs132774), *XRCC5* (rs207906), *PRKDC* (rs7003908), and *XRCC3* (rs861539) was assessed using TaqMan^®^ SNP Genotyping Assays and the TaqMan Universal Master Mix II, No UNG (Applied Biosystems, Foster City, CA, USA). The 20 µL total volume of the PCR reaction contained: 4 µL 5× HOT FIREPol^®^ Probe qPCR Mix (Solis, Tartu, Estonia), 1 µL 20× TaqMan SNP primers, 1 µL DNA (100 ng), and 14 µL RNA-free water. PCR reactions were performed under the following conditions: polymerase activation (10 min, 95 °C), 30 cycles of denaturation (15 s, 95 °C, 30) and hybridization/extension (60 s, 60 °C). Genotype analysis was performed on the Bio-Rad CFX96 system (BioRad, Hercules, CA, USA).

### 3.6. RNA Expression of DSB Repair-Related Genes

We performed Taq Man gene expression assay (Thermo Fisher Scientific Inc., Waltham, MA, USA) to assess the expression profile of seven genes associated with DSB, respectively: *BRCA2* (Hs01037416_m1), *BRCA1* (Hs02387156_m1), *ATM* (Hs01112362_m1), *RAD51B* (Hs01568761_m1), *RAD51* (Hs00947967_m1), *PRKDC* (Hs01016091_g1), and *H2AFX* (Hs01573336_s1). We used *RPLP1* (Hs02926887_g1) as a reference gene. The 10 µL total reaction volume was 10 µL including: 1 µL cDNA, 1 µL Primers, 5 µL 2× TaqMan^®^Universal Master Mix II, No UNG and 3 µL Nuclease free water. The conditions for the reaction were prepared according to the manufacturer’s protocol for the TaqMan^®^Universal Master Mix II, no UNG: polymerase activation (10 min, 95 °C), 30 cycles of denaturation (15 s, 95 °C), and annealing/extension (60 s, 60 °C). The qPCR reaction was performed in the Bio-Rad CFX96 system (BioRad, Hercules, CA, USA). Gene expression was calculated in relation to that of the reference genes (ΔCt sample = Ct target gene − Ct reference gene). Following, the relative mRNA expression was evaluated as a fold = 2^−ΔCt^ sample. Expression status was determined from the data regarding the expression value of the analyzed gene in the control patient population.

### 3.7. Statistical Analysis

The normal distribution of continuous variables was analyzed using the Shapiro–Wilk test. Descriptive data are expressed as median ± range according to lack of normal distribution. For the comparison of two groups (RA patients and healthy controls) the U Mann–Whitney rank sum test was used. The Hodges–Lehmann method was used to estimate the location shift. Multinomial logistic regression analyses were performed to calculate odds ratios (ORs) and 95% confidence intervals (CIs) for the effects of DNA repair status and other variables on RA. All variables included in the final multivariate models were determined to be independent by assessing their collinearity. Genotypes of DNA repair genes, were included as independent variables in univariate and multivariate multinomial logistic regression analyses. Only matching variables and factors that altered the ORs by 10% were included in the final multivariate models. The quality of the models was determined using the Hosmer–Lemeshow test. All statistical analyses were performed using TIBCO Statistica 13.3 (Palo Alto, CA, USA). In all tests, *p* value < 0.05 was used.

### 3.8. Decision Tree

A decision tree (DT) model is used in this study to recognize RA patients based on repair efficacy (RepEff) and genetic data, i.e., 29 SNP genes listed in Table 7. We define a sample representing a single patient or control as a tuple of 30 features, $x=(x_{1},\;\ldots ,\;x_{30})$, where $x_{1}$ is RepEff and $x_{2},\;\ldots ,\;x_{30}$ correspond to the successive SNP genes. All the features are treated as categorical. We consider N=90 samples of two equinumerous classes: RA and healthy. The output of the tree can is y=0 (healthy) or y=1 (RA).

The DT classifier is a powerful yet conceptually simple non-parametric algorithm that effectively partitions the feature space into smaller regions by recursively selecting the feature that offers the maximum information gain at each node. One of the key advantages of DT is its interpretability, allowing us to understand the underlying decision-making process. Moreover, this algorithm is versatile, capable of handling both numerical and categorical data, making it suitable for a wide range of applications.

We utilize the CART (classification and regression tree) implementation of the DT algorithm, as proposed by Breiman. The process of constructing a tree involves recursively partitioning a dataset into subsets based on the values of features. At each node, the algorithm selects the best feature to split the data, employing an impurity measure like the Gini index or entropy. This selection process is repeated recursively until a stopping criterion is met, such as reaching a predefined maximum number of splits. The resulting tree can then be utilized to predict the class of new samples by traversing from the root node to a leaf node corresponding to the predicted class. DT can be easily interpreted as a collection of simple rules that humans can readily understand.

The final decision of the DT classifier can be expressed as follows:$$\hat{y}=\sum_{l\in L}label\left(l\right)I(x\in l)$$
where L represents the set of leaves, function $label\left(l\right)$ assigns a label to a leaf l based on the subset of samples that reached that leaf, and function $I\left(x\in l\right)$ returns 1 when sample x reaches the specific leaf l, and 0 otherwise.

Typically, the label assigned by function $label\left(l\right)$ is the majority class within the subset that reached leaf l. The tests performed on the features in each node in our case has the following form: $x_{j}=v_{j}$, where j is the feature index selected individually for each node, and $v_{j}$ is a value selected from the domain of the j-th feature (e.g., A/A).

## 4. Discussion

This study appears to be the first to find a correlation between the less efficient repair of DSBs in PBMCs from patients with RA and SNPs within the *RAD51* and *RAD51B* genes. The results on less efficient DSB repair are consistent with others [9,10,11] and appear in RA, and higher endogenous DSB levels were also found in PBMCs isolated from patients with systemic sclerosis and systemic lupus erythematosus (SLE) [14,15]. Human cells repair DSBs by nonhomologous end joining (NHEJ) or homologous recombination (HR) and their variants [16]. The selection of the appropriate system depends on many factors and is initiated by DNA end resection that creates single-stranded DNA overhangs. The initiation of the HR system is dependent on RAD51 and its paralogs blocking the availability of DNA overhangs for NHEJ proteins [17]. RAD51B also has a different role in the repair of DSBs. It transduces the signal about DSBs to effector kinases like ATM, ATR, or DNA-PK and promotes the repair process [18]. *RAD51B* polymorphisms were associated with rheumatoid arthritis and erosion in patients with rheumatoid arthritis patients [19,20]. It seems that the allelic diversity of *RAD51* and *RAD51B* may influence their role in the repair of DSBs and contribute to less efficient repair. We have also shown that *RAD51B* expression is elevated in PBMCs isolated from subjects with RA. RAD51B interacts with other RAD51 paralogs to form a complex whose role in DSB repair is to stabilize protein foci in DNA overhangs. Given our limited knowledge of how RAD51 paralogs function, it is difficult to definitively determine the potential mechanism underlying the decreased efficiency of DSB repair when *RAD51B* is overexpressed. However, increased levels of one paralog can alter the delicate balance of the RAD51 paralog complex and contribute to the destabilization of repair protein foci. We also observed the overexpression of *BRCA1* and *BRCA2*. Both are known tumor suppressor genes, and the overexpression of some transcription variants is connected with delayed DSB repair [21]. In contrast to earlier findings by Shao et al. [9], these results suggest that *ATM* expression is not down-regulated in RA. There may be two reasons for this discrepancy. First, Shao et al. [9] analyzed the expression only in T lymphocytes. Second, they had a completely different ethnic composition of the study group, where more than 70% of the subjects were African American.

There are two potential hypotheses that explain the disruption of repair processes in RA. The first is related to chronic inflammation and associated oxidative stress [22]. Oxidative stress causes an increase in reactive oxygen species in the cell and increases the number of mutations in repair genes, for example, *TP53* [23], resulting in decreased repair efficiency. The second hypothesis is related to a faster immune ageing process in RA [24]. This results in the classic ageing T cell phenotype, which exhibits all the features of ageing metabolism, including reduced DNA repair efficiency. The two hypotheses are not mutually exclusive. They are coupled by a process called inflammaging, when ageing is accompanied by low-grade chronic inflammation [25]. Our results perfectly fit both hypotheses, since genetic background understood by the presence of allelic forms of DNA repair genes can stimulate both the ageing process (through inefficient DNA repair) and increase the mutation rate through deficiency in repairing oxidative DNA lesions as a consequence of chronic stress. Another possible explanation for the potential impact of inefficient double-strand break repair caused by the abnormal expression of DNA repair genes and their genetic variation on the development of RA is the generation of diversity in adaptive immunity known as V(D)J recombination [26]. The NHEJ system and the RAD51 protein are involved in this process [27,28]. In summary, it involves the introduction of DNA breaks within the receptor and immunoglubin genes and their unfaithful repair. Errors in V(D)J recombination can lead to induction of genotoxic stress and accelerate the ageing of T cells. T cell ageing is now recognised as a risk determinant in autoimmune syndromes, including RA and is associated with the progression of RA [29].

## 5. Conclusions

The results of this study suggest that genetic variations in *RAD51* and *RAD51B* genes contribute to the delayed/marginally efficient DSB repair phenotype in RA. We based our conclusion on logistic regression and confirmed it using machine learning and decision trees. Our study has some limitations. Although the SBS method identified four candidates in which SNPs can modulate DNA repair processes, in our opinion, more repetitive studies should be conducted on a larger study sample to confirm these relationships. Furthermore, functional work on the *RAD51* SNP should be performed. It is important to functionally characterize the genetic variants of *RAD51* and to find the biological mechanisms underlying the associations to assess the *RAD51* SNP as a prognostic and/or predictive biomarker in RA. Potential clinical applications include helping to identify an individual’s risk of RA and adjusting treatment for RA based on each patient’s unique molecular profile. In our opinion, profiling patients with RA for DNA repair deficiency is particularly important for the earlier detection of individuals who may be at risk of developing lymphomas as a consequence of RA and deficient DNA repair. For this reason, future studies should cover a broader time frame taking into account the DNA repair profile of RA patients and follow them in terms of any cancer occurrence. Further studies should also be conducted on other than Caucasian ethnic groups to answer the question of whether the phenotype–genotype relationships we have found are a common feature of the human race or whether they are limited to the Caucasian ethnic group.

## Figures and Tables

**Figure 1 ijms-25-02619-f001:**
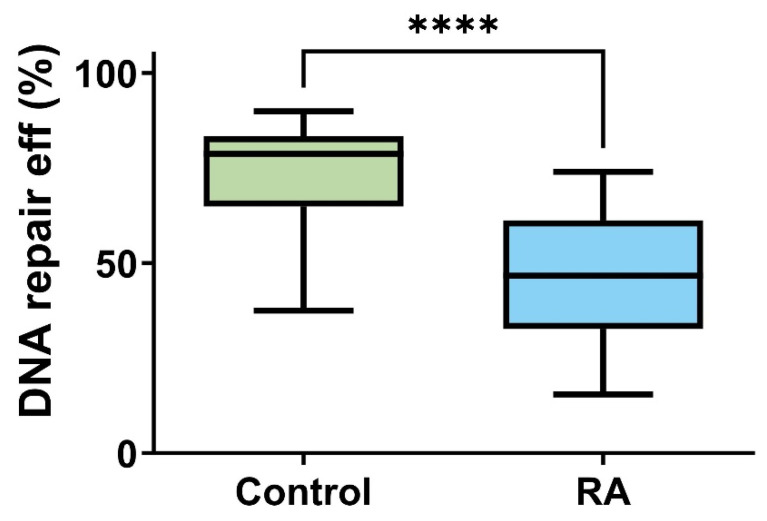
Distribution of individual DNA repair lesions induced by bleomycin efficiency in peripheral blood mononuclear cells (PBMCs) isolated from 45 healthy controls (green) and 45 rheumatoid arthritis (RA, blue) patients. Data are presented as medians. Differences between groups were analyzed using the U Mann–Whitney rank sum test analysis, **** means *p* < 0.0001.

**Figure 2 ijms-25-02619-f002:**
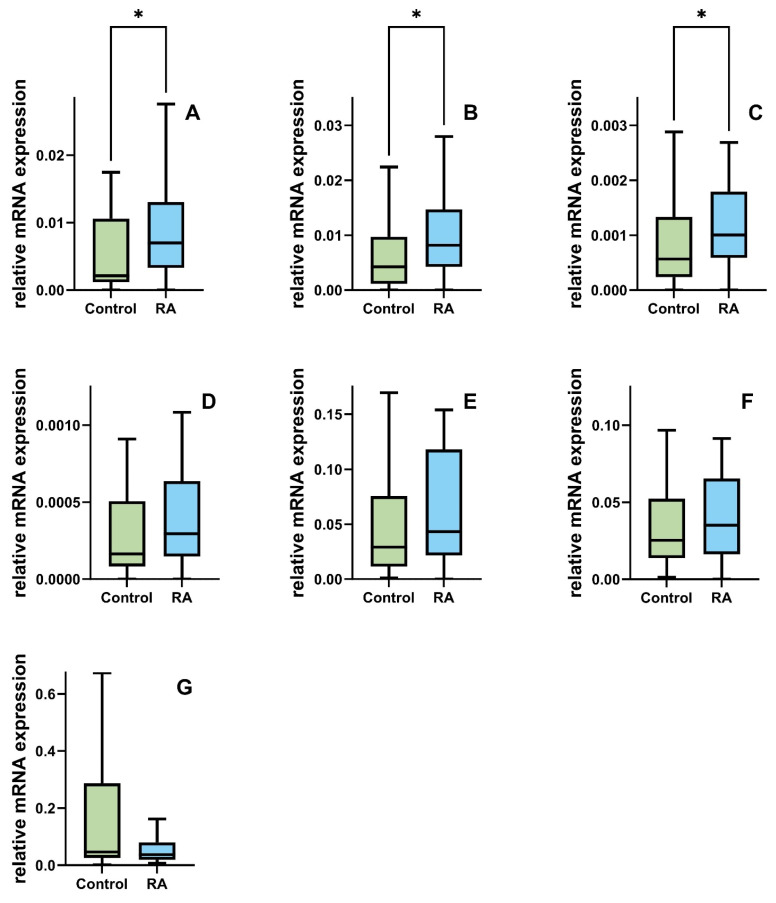
Comparison of the expression levels in peripheral blood mononuclear cells (PBMCs) of the key DNA double strands break repair genes (**A**) *RAD51B*, (**B**) *BRCA1*, (**C**) *BRCA2*, (**D**) *RAD51*, (**E**) *ATM*, (**F**) *PRKDC*, and (**G**) γ -*H2AX*) between the controls (green) *n* = 45 and patients with rheumatoid arthritis (blue) *n* = 45. Data are presented as medians. Differences between the groups were analyzed using the Mann–Whitney rank sum test analysis, * means *p* < 0.05.

**Figure 3 ijms-25-02619-f003:**
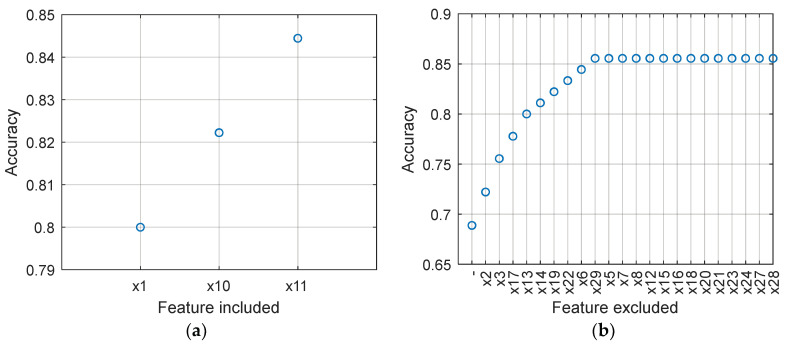
Accuracy of the DT classifier in the successive steps of SFS ((**a**) left panel) and SBS ((**b**) right panel).

**Figure 4 ijms-25-02619-f004:**
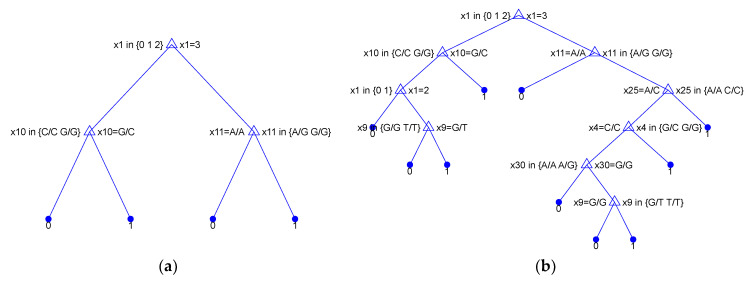
DT classifier built using features selected using SFS ((**a**) left panel) and SBS ((**b**) right panel).

**Figure 5 ijms-25-02619-f005:**
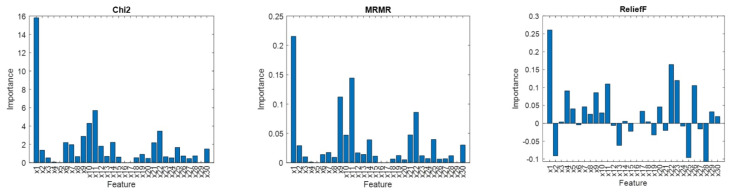
Feature importance.

**Table 1 ijms-25-02619-t001:** Characteristics of the study population.

Parameters	RA Patients *n* = 45	Control Group *n* = 45
sex	F-32 M-13	F-37 M-8
age	58.27 ± 13.45 years	54.73 ± 15.50
smoking	10	2
disease duration	14.55 ± 13.12 years	
remission	yes 4; no 41	
CRP	18.26 ± 22.02	
ESR	28.33 ± 23.20	
RF	220.38 ± 333.60	
ACPA	1169.36 ± 4291.60	
treatment	Methotrexate 17/45	
	Sulfasalazine 4/45	
	No DMARDs 22/45	
	Glucocorticosteroids 22/45	

CRP: C-reactive protein; ESR: erythrocyte sedimentation rate; RF: rheumatoid factor; ACPA: anti-citrullinated peptide antibodies; GCS: glucocorticosteroids; F: female; M: male.

**Table 2 ijms-25-02619-t002:** Efficiency of the repair of DNA lesions (RepEff) that induced by bleomycin in peripheral blood mononuclear cells (PBMCs) isolated from 45 healthy controls and 45 rheumatoid arthritis (RA) patients. The efficiency was calculated as follows: The DNA damage measured immediately after exposure to bleomycin was set as 100% of DNA damage. Next, the percentage of the repaired DNA after 120 min was measured.

RepEff	Percentage of Total Repair	Controls (*n* = 45)	RA (*n* = 45)
Group 1 Highly efficient DNA repair	>83.4	12 (26.8%)	1 (2.2%)
Group 2 Efficient DNA repair	78.7–83.3	11 (24.4%)	1 (2.2%)
Group 3 Marginally efficient DNA repair	65.1–78.6	11 (24.4%)	5 (11.1%)
Group 4 No DNA repair	<65	11 (24.4%)	38 (84.5%)

**Table 3 ijms-25-02619-t003:** The allele and genotype frequency of the common polymorphisms located in key DNA double-strand break repair-related genes in control and rheumatoid arthritis (RA) groups.

Polymorphism/Gene	Genotype	Control	RA	OR (95% CI)
rs3218536/*XRCC2*	C/C	3 (6.7%)	1 (2.2%)	1.00
	C/T	41 (91.1%)	43 (95.6%)	3.15 (0.31–31.49)
	T/T	1 (2.2%)	1 (2.2%)	3.00 (0.08–107.45)
rs1801320/*RAD51*	G/G	32 (71.1%)	19 (40.0%)	1.00
	G/C	2 (4.4%)	8 (17.8%)	**6.74 (1.29–35.08)**
	C/C	11 (24.4%)	18 (42.2%)	**2.75 (1.07–7.06)**
rs7180135/*RAD51*	G/G	5 (11.1%)	4 (8.9%)	1.00
	A/G	22 (48.9%)	31 (68.9%)	1.76 (0.42–7.31)
	A/A	18 (40%)	10 (22.2%)	0.69 (0.15–3.19)
rs45549040/*RAD51*	A/A	41 (91.1%)	43 (95.6%)	1.00
	A/C	3 (6.7%)	0 (0%)	0.00 (0.00–NA)
	C/C	1 (2.2%)	2 (4.4%)	1.91 (0.17–21.84)
rs1801321/*RAD51*	G/G	17 (37.8%)	7 (15.6%)	1.00
	G/T	9 (20%)	11 (24.4%)	2.97 (0.85–10.31)
	T/T	19 (42.2%)	27 (60%)	**3.45 (1.20–9.94)**
rs2619681/*RAD51*	C/C	30 (66.7%)	30 (66.7%)	1.00
	C/T	13 (28.9%)	13 (28.9%)	1.00 (0.40–2.51)
	T/T	2 (4.4%)	2 (4.4%)	1.00 (0.13–7.57)
rs963917/*RAD51B*	G/G	12 (26.7%)	26 (57.8%)	1.00
	A/G	23 (51.1%)	17 (37.8%)	**0.34 (0.13–0.86)**
	A/A	10 (22.2%)	2 (4.4%)	**0.09 (0.02–0.49)**
rs963918/*RAD51B*	C/C	7 (15.6%)	5 (11.1%)	1.00
	C/T	17 (37.8%)	26 (57.8%)	2.14 (0.58–7.86)
	T/T	21 (46.7%)	14 (31.1%)	0.93 (0.25–3.54)
rs3784099/*RAD51B*	G/G	30 (66.7%)	20 (44.4%)	1.00
	A/G	11 (24.4%)	23 (51.1%)	**3.14 (1.26–7.83)**
	A/A	4 (8.9%)	2 (4.4%)	0.75 (0.13–4.49)
rs10483813/*RAD51B*	T/T	34 (75.6%)	33 (73.3%)	1.00
	A/T	8 (17.8%)	11 (24.4%)	1.42 (0.51–3.96)
	A/A	3 (6.7%)	1 (2.2%)	0.34 (0.03–3.47)
rs1042522/*TP53*	C/C	21 (46.7%)	26 (57.8%)	1.00
	C/G	11 (24.4%)	14 (31.1%)	1.03 (0.39–2.73)
	G/G	13 (28.9%)	5 (11.1%)	0.31 (0.10–1.01)
rs1051669/*RAD52*	C/C	31 (68.9%)	25 (55.6%)	1.00
	C/T	14 (31.1%)	19 (42.2%)	1.68 (0.71–4.01)
	T/T	0 (0%)	1 (2.2%)	NA (0.00-NA)
rs2155209/*MRE11A*	T/T	15 (33.3%)	15 (33.3%)	1.00
	C/T	19 (42.2%)	26 (57.8%)	1.37 (0.54–3.46)
	C/C	11 (24.4%)	4 (8.9%)	0.36 (0.09–1.40)
rs132774/*XRCC6*	G/G	20 (44.4%)	16 (35.6%)	1.00
	C/G	13 (28.9%)	18 (40%)	1.73 (0.66–4.57)
	C/C	12 (26.7%)	11 (24.4%)	1.15 (0.40–3.27)
rs207906/*XRCC5*	G/G	33 (73.3%)	29 (64.4%)	1.00
	A/G	8 (17.8%)	12 (26.7%)	1.71 (0.61–4.75)
	A/A	4 (8.9%)	4 (8.9%)	1.14 (0.26–4.96)
rs7003908/*PRKDC*	A/A	14 (31.1%)	10 (22.2%)	1.00
	A/C	27 (60%)	25 (55.6%)	1.30 (0.49–3.44)
	C/C	4 (8.9%)	10 (22.2%)	3.50 (0.85–14.41)
rs861539/*XRCC3*	G/G	33 (73.3%)	39 (86.7%)	1.00
	A/G	4 (8.9%)	1 (2.2%)	0.21 (0.02–1.99)
	A/A	8 (17.8%)	5 (11.1%)	0.53 (0.16–1.77)

Bold indicates statistically significant results; NA—not determined.

**Table 4 ijms-25-02619-t004:** Multivariate logistic regression analysis of the efficiency of the repair of DNA lesions induced by bleomycin and common polymorphisms located in DNA double-strand break repair-related genes in the control and rheumatoid arthritis (RA) groups.

Factor	Adjusted for	OR (95% CI)
RepEff	N/A	41.4 (4.8–355.01)
	rs1801320	42.52 (4.4–408.2)
	rs1801321	44.93 (5.05–399.83)
	rs963917	69.7 (5.9–816.8)
	rs10483813	57.4 (4.7–698.5)
	rs1801321 and rs1801320 (*RAD51*)	53.5 (4.7–613.21)
	rs963917 and rs3784099 (*RAD51B*)	73.4 (5.3–1011.05)

**Table 5 ijms-25-02619-t005:** SNP features selected using SFS and SBS.

SNP Features Selected by SFS		SNP Features Selected by SBS	
$x_{1}$	RepEff	$x_{1}$	RepEff
$x_{10}$	RAD51_rs1801320-RNG	$x_{4}$	OGG1_rs1052133-RNG
$x_{11}$	RAD51B_rs963917-RNG	$x_{9}$	RAD51_rs1801321-RNG
		$x_{10}$	RAD51_rs1801320-RNG
		$x_{11}$	RAD51B_rs963917-RNG
		$x_{25}$	PRKDC_rs7003908-RNG
		$x_{26}$	TDG_rs4135054-RNG
		$x_{30}$	XRCC3_rs861539-RNG

**Table 6 ijms-25-02619-t006:** Performance of DT with different input configurations.

DT Inputs	TP	FP	FN	TN	Accuracy	Precision	Sensitivity	Specificity	F1 Score
$x_{1}$	38	7	11	34	0.8000	0.8444	0.7755	0.8293	0.8085
$x_{1},\;\ldots ,\;x_{30}$	27	18	10	35	0.6889	0.6000	0.7297	0.6604	0.6585
Selected by SFS	39	6	8	37	0.8444	0.8667	0.8298	0.8605	0.8478
Selected by SBS	37	8	5	40	0.8556	0.8222	0.8810	0.8333	0.8506

**Table 7 ijms-25-02619-t007:** Single-nucleotide polymorphisms analyzed in this study.

SNP (Gene Name)	Pathway	Chromosome	Positions	Allele	Minor Allele Frequency	
					Case	Control
rs3218536 (*XRCC2*)	DSB	7	152648922	C/T	0.5	0.48
rs1801320 (*RAD51*)	HR	15	40695330	G/C	0.49	0.33
rs7180135 (*RAD51*)	HR	15	40731896	A/G	0.43	0.36
rs45549040 (*RAD51*)	HR	15	40732091	A/C	0.04	0.06
rs1801321 (*RAD51*)	HR	15	40695367	G/T	0.28	0.48
rs2619681 (*RAD51*)	HR	15	40696823	C/T	0.19	0.19
rs963917 (*RAD51B*)	HR	14	68595606	A/G	0.23	0.48
rs963918 (*RAD51B*)	HR	14	68595397	C/T	0.4	0.34
rs3784099 (*RAD51B*)	HR	14	68283210	A/G	0.3	0.21
rs10483813 (*RAD51B*)	HR	14	68564567	A/T	0.14	0.16
rs1042522 (*TP53*)	DSB	17	7676154	C/G	0.27	0.41
rs1051669 (*RAD52*)	DSB	12	913286	C/T	0.23	0.16
rs2155209 (*MRE11A*)	DSB	11	94417624	C/T	0.38	0.46
rs132774 (*XRCC6*)	NHEJ	22	41635949	C/G	0.44	0.41
rs207906 (*XRCC5*)	NHEJ	2	216148178	A/G	0.22	0.18
rs7003908 (*PRKDC*)	NHEJ	8	47858141	A/C	0.5	0.39
rs861539 (*XRCC3*)	HR	14	103699416	A/G	0.12	0.22

DSB Repair: double-strand break repair; HR: homologous recombination: NHEJ: non-homologous end joining.

## Data Availability

The data presented in this study are available on request from the corresponding author. The data are not publicly available due to regulations in the country of the correspondence author.

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
