# Peer review of "The Association between Inefficient Repair of DNA Double-Strand Breaks and Common Polymorphisms of the HRR and NHEJ Repair Genes in Patients with Rheumatoid Arthritis"

_ijms, 2024, doi:10.3390/ijms25052619_

Round 1

Reviewer 1 Report

Comments and Suggestions for Authors

The  study investigates the association between DNA repair inefficiency and polymorphisms in repair genes in patients with Rheumatoid Arthritis (RA). The paper details an experiment comparing DNA repair capabilities in RA patients to healthy controls using a comet assay. The authors also look at the expression levels of DNA repair genes and single nucleotide polymorphisms (SNPs) associated with RA.

Here are my comments:

Strengths of the paper include a clear methodology for the comet assay, a well-defined patient cohort, and the use of multivariate logistic regression to analyze the influence of SNPs on DNA repair efficiency. Moreover, the use of decision tree analysis adds robustness to the findings.

However it would better to improve as follows:

Areas to strengthen could include more detailed discussion on the implications of the identified gene expressions and SNPs on RA pathology and potential therapeutic targets. It would also be beneficial if the study could expand on how these genetic variations contribute to the overall disease mechanism in RA. There could be a deeper exploration of the limitations and how they might impact the interpretation of the results, as well as a broader consideration of how these findings fit within the existing body of literature.

Please add more clearly details about Ethic: Does the study meet ethical standards for research?

Reviewer 2 Report

Comments and Suggestions for Authors

ijms-2820887

The association between inefficient repair of DNA double strand breaks and common polymorphisms of the HRR and NHEJ repair genes in patients with Rheumatoid Arthritis

This research article focuses on the correlation of reduced DNA double-strand break (DSB) repair efficiency in rheumatoid arthritis (RA) patients with single nucleotide polymorphisms (SNPs) in DSB repair genes. The authors report that peripheral blood mononuclear cells derived from RA patients were characterized by significantly reduced repair of bleomycin-induced DNA lesions in comparison to healthy controls. Further analyses suggest that specific SNPs in certain repair genes (esp. rs1801320 in RAD51 and rs963917 in RAD51B) may play a role in this context.

The study is well designed, carried out properly, and technically sound. The results are comprehensible and conclusive. Text and figures are straightforward and clear. However, there are some critical points/questions requiring the authors’ consideration.

Major point:

 1.     Results/3.1: Since the disease duration strongly differs among RA patients, please include the information whether the repair efficiency correlates with the age and/or disease duration of the patients or the age of the healthy controls.

Minor points:

2.     The text of the manuscript has to be revised (check for missing/redundant spaces, punctuation, singular/plural forms, Greek characters, superscripted characters, …).

3.     All abbreviations have to be defined in the text (including figure/table legends) when they are used for the first time and should be used afterwards throughout the manuscript.

4.     Materials and Methods/2.2: Please indicate the amount of blood used, the number of PBMC isolated, and the purity of the isolated PBMC population.

5.     Materials and Methods/2.3: In line 92, a treatment of PBMC with 25 M bleomycin is indicated. Is this correct?

6.     Materials and Methods/2.5 and 2.6: I suggest rephrasing the description of the PCR conditions (in its present form, it looks as if 30 cycles of denaturation were followed by 30 cycles of hybridization/extension).

7.     Table 1: Please adjust the defined abbreviations provided at the end of the table to the abbreviations used in the table.

8.     Results/3.1: Please provide detailed data on age, sex, smoking, clinical data, and medication in both groups (e.g., in a table).

9.     Table 4: It should be mentioned that the ORs provided in Table 4 refer to the comparison of the forth with the first group.

10.  Results/3.5, lines 265-266: For a better overview, please include the information which group shows higher expression levels.

11.  Figure 2: Please include the numbers of analyzed samples in the figure legend.

12.  Discussion: The (potential) impact of the relevant SNPs analyzed on the respective genes (e.g., the effect on expression, function, or similar) and on RA development should be discussed.

Comments on the Quality of English Language

The text of the manuscript has to be revised (check for missing/redundant spaces, punctuation, singular/plural forms, Greek characters, superscripted characters, …).

Round 2

Reviewer 2 Report

Comments and Suggestions for Authors

ijms-2820887

The manuscript provides a revised version of the study “The association between inefficient repair of DNA double strand breaks and common polymorphisms of the HRR and NHEJ repair genes in patients with Rheumatoid Arthritis”. The manuscript has been improved and my comments have been adequately addressed.